# Research on the Impact of Cooperative Membership on Forest Farmer Household Income and Assets—Case Study from Liaoning Herbal Medicine Planting Cooperatives, China

**Jingyu Wang** [1,*], **Zhe Zhao** [2] **and Lei Gao** [3]

1    School of Economic and Management, Zhejiang A&F University, Hangzhou 311300, China
2    School of Economics, Faculty of Economics, Liaoning University, Shenyang 110136, China; zhaozhe@lnu.edu.cn
3    School of Economics and Management, Yanshan University, Qinhuangdao 066004, China; gaolei@ysu.edu.cn
*    Correspondence: wangjy@zafu.edu.cn; Tel.: +86-182-1057-9605

**Abstract:** Improving the income and assets of forest farmers is the basis for realizing the sustainable development of forestry. In this paper, we tested the impact of membership in herbal medicine planting cooperatives on forest farmer household income and assets using the propensity score matching (PSM) method and household surveys of the study area. The results showed that cooperative membership can greatly improve forest farmer household income and assets; the higher the accumulation of forest farmer household social capital and human capital, the more farmers were inclined to participate in cooperatives. Householders who were migrant workers were more likely to make the decision to participate in cooperatives compared with those without migrant work experiences. The results of ATT further verified the conclusion that cooperative membership can significantly improve income and assets, which increased by 7.04% and 4.19%, respectively. In addition, according to the survey, the current development of cooperatives in the forestry area experienced problems such as inconsistent quantitative and qualitative development, insufficient driving force, irregular operating mode, inaccurate policy support, and inadequate guidance mechanisms. This paper focused on innovating cooperation mechanisms, enriching joint forms, enhancing driving capacity, stimulating internal driving forces, strengthening system construction, improving standards, enhancing guidance services, and strengthening institutional guarantees. These recommendations have been put forward to guide policy for sustainable forest development.

**Keywords:** forest farmer; specialized cooperatives; farmer household income; farmer household assets; PSM model





## 1. Introduction

Specialized farmer cooperatives (hereafter, "cooperatives") are cooperative economic organizations of mutual support and democratic management on the basis of household contract management. These cooperatives not only overcome the contradiction between "small production" in the small-scale peasant economy and "large demand" in the market economy, but also meet consumer requirements for the quality and safety of agricultural products. They represent an important organizational means of realizing the organic connection between small farmers and modern agricultural development [1–3]. Considering the basic conditions of "big country and smallholders," smallholder management will remain the basis of China's agriculture [4]. Cooperatives not only allow farmers to participate in agricultural production and professional cooperation, but also allow them to benefit by effectively connecting scattered individual farmers with agricultural modernization. In this way, cooperatives make a significant contribution to farmers by promoting agricultural modernization [5–7]. Consequently, the government has actively taken various measures to

support the development of cooperatives and promote economic cooperation and market integration [8].

From the perspective of national legislation, the Agricultural Law of the People's Republic of China, which came into effect in March 2003, stipulated that "farmers are encouraged to voluntarily form all kinds of professional cooperative economic organizations on the basis of household contract operations." According to the Law of the People's Republic of China on Specialized Farmer Cooperatives, "the State shall promote the development of farmer specialized cooperatives through financial support, preferential taxation, support for finance, science and technology, personnel, and guidance of industrial policies and other measures." According to the Rural Revitalization Promotion Law of the People's Republic of China, "the State supports farmer specialized cooperatives, family farms, agriculture-related enterprises, e-commerce enterprises, and specialized agricultural social service organizations to establish a close interest linkage mechanism with farmers in a variety of ways, so that farmers can share the value-added benefits of the whole industrial chain." On this basis, the government has also formulated specific policies to support the development of cooperatives, especially providing important financial support [9,10]. In terms of allocating public budget funds, the central government generally allocates funds to provincial governments to meet local needs to support the development of cooperatives [11]. This practice has laid a financial foundation for the development of cooperatives in various regions. In addition, local governments below the provincial level also often increase financial support for cooperatives from their budgets.

In recent years, with strong support from government departments at all levels, cooperatives have grown rapidly, totaling 2.2 million nationwide by 2020. However, whether the development of cooperatives is conducive to the realization of relevant government goals (such as increasing farmers' income, promoting the effective connection between small farmers and modern agriculture, narrowing the income gap, etc.) has become the focus of attention of scholars and government departments. A large number of existing studies have shown that the impact of cooperative membership on farmers' household income is significantly positive [12–14]. Cooperatives and enterprises are important market suppliers of agricultural socialized services, and it is likely that the agricultural socialized services market will gradually form a multi-subject competitive supply pattern. As the strength of cooperatives increases, they can compete for the market share of enterprises and improve the welfare of farmers through lower prices and higher utility effects [15]. At the same time, they force enterprises to reduce the price of production means, improve the welfare of participating farmers, and contribute to the improvement of the overall welfare of farmers. Furthermore, many studies consider that the increased income derived from cooperatives differs for different types of households. Some researchers believe that this effect is more obvious for large-scale and high-income households [16], while others believe that the effect is more obvious for low-income and poor households [17]. Yet others consider that the efficiency of cooperatives is relatively low and the service function needs to be strengthened [18].

Overall, further research is still necessary to examine the following aspects in greater depth. Firstly, most of the existing research has focused on the impact of cooperative membership on household income; however, it has ignored the impact on household assets, which also form an important part of family welfare. Secondly, most of the existing research has focused on the macro level and the theoretical level. Therefore, there is a lack of a practical approach to examine specific regions, especially quantitative analysis based on survey data. Due to differences in cooperative types, industrial characteristics, regional resource endowments, socioeconomic development levels, research perspectives, and other factors, the conclusions drawn will inevitably be different. Therefore, it is difficult to determine a development path that is suitable for the actual situation of specific regions. Thirdly, most of the existing research has focused on the whole industry, and studies are lacking research on the characteristic industrial cooperatives that rely on regional resource endowment, which has played a huge role in industrial poverty alleviation.

This paper selected the farmer households that plant herbal medicine in the eastern mountainous forest area of Liaoning province as research object. We empirically tested the impact of membership in herbal medicine planting cooperatives on forest farmer household income and assets through the combination of theory and demonstration. We also explored the potential impact of cooperatives on continuously promoting increasing household income and guaranteeing the stable and sustainable development of forestry area, in order to obtain new supporting evidence.

## 2. Theoretical Analysis

### 2.1. The Impact Mechanism of Cooperative Membership on Farmer Household Income

Firstly, economies of scale come from collective action. Compared with investor-owned enterprises, cooperatives are more conducive to saving transaction costs by trading with farmers [19,20]. The central position of cooperatives in the agricultural organization system is determined by the collective action of farmers through cooperatives in an ideal way. Moreover, rural elites with Party member farmers as the core play a leading and exemplary role; they organize ordinary farmers through brand building, deep processing, market negotiation, signing orders, unified purchase of agricultural products, cooperative inspection, coordinated sales, unified management, and other ways, to achieve the high added value of agricultural products that are difficult for individual farmers to achieve and can improve the level of farmer household income [21]. The core members of cooperatives rely on the contribution of ordinary farmers to the quantity and quality of products so that they can achieve a scale effect in quantity and a value-added impact on quality, to save transaction costs and obtain corresponding bargaining power in the market [22]. Furthermore, cooperatives with strong profitability have great development potential. They will have obvious driving effects and will continue to follow the path of specialized joint development and organize family farms and cooperatives in villages and towns to form joint cooperatives, which will further benefit from economies of scale [23]. Cooperatives actively promote the organic integration of agricultural production, the processing and marketing of agricultural products, catering, leisure, rural culture, tourism, education, fitness, and other emerging industries, to extend the agricultural industrial chain, realize the integrated development of three sectors, drive increases in the village's collective income, and help farmers get rich [24].

Secondly, contract scales, standardization, and brand development construction also have an obvious effect on increasing income. Cooperatives sell their products in bulk through sales contracts signed before harvest, reducing the market risk of price fluctuations of agricultural products [25]. This benefit has been fully demonstrated by cooperatives in China and many other countries. Cooperatives that seek to maximize the welfare of their members have more incentive than investor-owned cooperatives to invest in innovations aimed at improving quality, thereby improving the nature of product differentiation and market structure. Branding the agricultural products of cooperatives can make them more distinctive, and the prices will be higher. Cooperative members can earn 20% more than non-member farmers by promoting the standardization and branding of agricultural products in China [26].

Thirdly, according to the provisions of the Law of Cooperatives, the surplus distribution of the cooperative is based on the agreement between the members and the cooperatives, and the distributable surplus is mainly returned in line with the proportion of the trading volume (amount) between the members and the cooperatives, and not less than 60% of the total amount. After the return, the remaining part is evenly quantified to the members based on the amount of capital contribution recorded in the members' accounts and the share of the provident fund, as well as the property formed by the direct subsidies from the state finance and donations from others, and is distributed to the members in proportion.

Finally, cooperatives have access to many kinds of government support. According to the relevant provisions of the Law of Cooperatives, the central and local financial de-

partments shall separately allocate funds to provide support for cooperatives to carry out such services as information, training, standards and certification of agricultural products (quality), construction of agricultural production infrastructure, marketing, and technology promotion [27]. And cooperatives enjoy tax concessions in agricultural production, processing, distribution, services, and many other agriculture-related economic activities [28].

### 2.2. The Impact Mechanism of Cooperative Membership on Farmer Household Assets

On the one hand, the advantages of mechanization and scale and the supporting role of the government's agricultural machinery purchase subsidy will encourage farmers to buy advanced and applicable agricultural machinery. Farmers can buy agricultural machinery supported and popularized by governments at subsidized prices and participate in cooperatives to carry out mechanized production with agricultural machinery. This can not only improve agricultural production efficiency but also reduce labor input, thereby liberating a large number of labor forces [29]. Additionally, by participating in cooperatives that provide large-scale, professional, and social services, farmers can not only earn service fee income but also obtain additional government subsidies. Increasing income can further encourage farmers to purchase agricultural machinery that is more advanced and applicable and ensure that they can afford to buy it [30].

On the other hand, the purchase price of agricultural machinery is generally high, and the government subsidy for farmers to buy agricultural machinery is usually less than 30%, which stimulates a large amount of capital investment by cooperative members (the funds come from the farmers' own funds and bank loans, etc.) [31]. Moreover, the local agricultural machinery purchase subsidy limits the quantity of agricultural machinery purchased by cooperatives more loosely, resulting in a higher asset effect.

## 3. Materials and Methodology

### 3.1. Variables

#### 3.1.1. Treatment Variable

This paper selected membership in herbal medicine planting cooperatives as a treatment variable, and a counterfactual hypothesis was adopted to analyze the treatment effect of cooperative membership on farmer household income and assets. Based on this, cooperative membership was set as a dummy variable. If the sample household participated in a cooperative, it was assigned a value of 1 (as the treatment group). If not, it was assigned a value of 0 (as the control group).

#### 3.1.2. Consequence Variable

According to the research objective, this paper selected farmer household income and farmer household assets as consequence variables from survey data. And the logarithm of the total income of the farmer household and the valuation of agricultural operating fixed assets owned by the farmer household were taken as specific indexes.

#### 3.1.3. Matching Variable

Based on the existing research, this paper selected household social capital, human capital, land area, labor scale, householder age, the education level of the householder, and the migrant work experience of the householder as matching variables [17,24,32]. The definitions of variables are shown in Table 1. The social capital (Table 2) and human capital of farmers were obtained by referring to the research of Liu et al. (2020) [33] and calculated by principal component factor analysis score.

**Table 1.** The definitions of variables.

| Variables | Definitions |
| --- | --- |
| Participating cooperatives (PC) | Participating (as treatment group) = 1; Non- participating (as control group) = 0. |
| Farmer household income (income) | The logarithm of the total income related to herbal medicine planting and processing of the farmer's household in 2021 (in CNY). |
| Farmer household assets (assets) | The logarithm of the valuation of agricultural operating fixed assets owned by the farmer's household in 2021 (in CNY). |
| Farmer household social capital (social) | Principal component factor analysis score. |
| Farmer household human capital (human) | Principal component factor analysis score. |
| Farmer household land area under operation (land) | The total area of land planted with herbal medicine by the farmer's household. |
| Farmer household labor scale (labor) | The number of labor forces participating in herbal medicine planting and processing. |
| Householder age (age) | - |
| The education level of the householder (education) | Primary school = 1; junior high school = 2; high school = 3; college = 4. |
| The migrant work experience of the householder (migrant) | Yes = 1; No = 2. |

**Table 2.** Composition of farmer household social capital and human capital.

| Variables | Dimension | Definitions |
| --- | --- | --- |
| Farmer household social capital (social) | Social network | Relationship with relatives and friends |
| | Social trust | Relationship with village leaders<br>The number of people you can trust in your village<br>The degree of trust in the village committee<br>Possibility of finding something lost in the village |
| | Social prestige | The importance of individual opinions in village collective decision making<br>Probability of others asking you for help when they are in trouble<br>Frequency of mentoring others<br>Frequency of resolving conflicts in the village |
| | Social participation | Willingness to participate in village collective activities |
| Farmer household human capital (human) | Knowledge | Average education level of the household workforce |
| | Capacity | Average age of the household workforce<br>Average health level of the household |

*3.2. Data*

The data used in this paper were all taken from a micro survey in Liaoning province. The research group designed the questionnaire according to the characteristics of agricultural development and agricultural resource endowment in Liaoning province and adjusted the questionnaire on the basis of pre-investigation. The micro survey was carried out in the eastern mountains in June 2022. During the survey process, the interviewers asked questions, and the farmers answered them. And the information was not filled in the questionnaire until the interviewers confirmed that it was correct and reviewed the questionnaire, thus ensuring the accuracy of the survey information. The survey covered farmers' operational expenses and income, operation and management, natural risks, market risks, family assets, financial services, and other aspects. After eliminating the questionnaires that omitted key information or provided contradictory information, 511 valid questionnaires were obtained, with an effective rate of 95.93%. In this paper, 248 farmers who grow herbal medicine in forest area were selected as the specific analysis objects. The basic information of the sample is shown in Table 3.

**Table 3.** Data description.

| | Sample N = 248 | | Treatment Group N = 71 | | Control Group N = 177 | |
|---|---|---|---|---|---|---|
| | **Mean** | **Standard** | **Mean** | **Standard** | **Mean** | **Standard** |
| *income* | 63,884 | 647,331 | 203,673 | 1204,276 | 7810 | 14,802 |
| *assets* | 330,227 | 3,224,564 | 1,013,752 | 6,000,990 | 56,044 | 73,455 |
| *social* | 0.573 | 0.173 | 0.651 | 0.166 | 0.541 | 0.166 |
| *human* | 0.501 | 0.163 | 0.562 | 0.147 | 0.477 | 0.163 |
| *land* | 57.218 | 503.425 | 157.359 | 936.444 | 17.048 | 35.053 |
| *labor* | 2.315 | 0.925 | 2.366 | 1.072 | 2.294 | 0.862 |
| *age* | 57.649 | 9.632 | 54.690 | 9.503 | 58.836 | 9.452 |
| *education* | 1.831 | 0.739 | 1.972 | 0.755 | 1.774 | 0.727 |

It can be seen from Table 3 that 71 households participated in cooperatives and 177 households did not. Without controlling for other socioeconomic characteristics, the between-group differences in farmer household income and farmer household assets for participating in cooperatives or not were significantly positive at a 1% level. Additionally, there were also significant differences in farmer household social capital, farmer household human capital, farmer household land area under operation, farmer household labor scale, householder age, the education level of the householder, and migrant work experience of the householder among the farmer households that participated in cooperatives or not. However, a simple comparison of means can only reflect the differences between participating and not participating in terms of correlation. To investigate the impact of cooperative membership on farmer household income and assets from the perspective of causality, a more rigorous econometric analysis method was needed.

*3.3. Methodology*

This paper selected the estimation method of propensity score matching (PSM) to quantitatively analyze the impact of cooperative membership on farmer household income and assets. The specific steps were as follows:

Firstly, we regarded whether the ith farmer household did or did not have a cooperative membership as a binary random variable. Where, $PC_i = 1$ represents the farmer with a cooperative membership; $PC_i = 0$ represents the farmer without a cooperative membership; $income_i$ and $assets_i$ represent the ith farmer household income and assets, respectively; $income_{1i}$ and $assets_{1i}$ represent the farmer household income and assets of $PC_i = 1$, $income_{0i}$ and $assets_{0i}$ represent the farmer household income and assets of $PC_i = 0$. Therefore, the change in farmer household income and assets resulting from cooperative membership can be represented by $income_{1i} - income_{0i}$ and $assets_{1i} - assets_{0i}$. As the farmer household income and assets of participating or non-participating cooperatives cannot be observed simultaneously, they can be defined as follows:

$$income_i = (1 - PC_i) \times income_{0i} + PC_i \times income_{1i} = income_{0i} + PC_i \times (income_{1i} - income_{0i})$$

$$assets_i = (1 - PC_i) \times assets_{0i} + PC_i \times assets_{1i} = assets_{0i} + PC_i \times (assets_{1i} - assets_{0i})$$

Secondly, as the farmer households were non-random and self-selecting when choosing whether to participate in cooperatives or not. Moreover, since the data of participating and non-participating cooperatives cannot be obtained at the same time, direct comparison easily produced endogeneity. The propensity score matching (PSM) model can effectively solve the problem of endogeneity through constructing a counterfactual hypothesis to reduce the multidimensional information of farmer households to a factor and match the farmers who participated in cooperatives or not in multiple dimensions. Specifically, in the set of farmer households that did not participate in cooperatives, one or some will be selected to match each farmer household that did participate in a cooperative. When the analysis sample was restricted to individuals that had already received "treatment",

the Average Treatment Effects on the Treated (ATT) was obtained, that is, the impact of cooperative membership on farmer household income and assets.

$$E(income1i - income_{0i}|PC_i = 1)$$

$$E(assets_{1i} - assets_{0i}|PC_i = 1)$$

Under the assumption of conditional expectation independence, the ATT can be estimated by the PSM model through matching the different farmer households with similar characteristics that participated or did not participate in cooperatives through estimating each individual's propensity score, namely, $P(X_i)$.

$$ATTincome = \frac{1}{N1} \sum_{i=1}^{N1} (income_{1i} - \sum_{j \in PC(pi)} wijincome_{0i})$$

$$ATTassets = \frac{1}{N1} \sum_{i=1}^{N1} (assets_{1i} - \sum_{j \in PC(pi)} wijassets_{0i})$$

where $N_1$ represents the number of individuals in the treatment group; $PC(pi)PC(pi)$ represents the paired group of the *i*th individual in the treatment group; $wijwij$ represents the weight of each individual in the paired group of the *i*th individual, and $\sum_{j \in PC(pi)} wij = 1 \sum_{j \in PC(pi)} wij = 1$. Moreover, $PC(pi)PC(pi)$ and $wijwij$ will definitely be different under different matching methods. On the basis of reference to existing research, the four PSM methods of Nearest Neighbor Matching (NNM), Local Linear Regression Matching (LLR), Kernel Matching (KM), and Radius Matching (RM) were selected to estimate the ATT in this paper.

## 4. Results

### 4.1. Baseline Regression Analysis

Tables 4 and 5 show the baseline regression results of farmer household income and assets. It can be seen that the coefficients of determination $R^2$ were both higher than 15%, indicating that the model had a high level of fitting, and the independent variables greatly explained the dependent variable, which can estimate the impact of cooperative membership on farmer household income and assets accurately. The coefficients of the variable PC were also both significantly positive at a 1% level, indicating that cooperative membership can greatly improve farmer household income and assets.

The variables of farmer household human capital and farmer household land area under operation passed the significance test, as seen in Tables 4 and 5, indicating that the accumulation of social capital and scale operation both helped to increase farmer household income and assets, and the research of Khan et al. (2022) supported this result [34]. Additionally, the migrant work experience of householders had a positive effect on farmer household income, while the farmer household labor scale had a positive effect on farmer household assets. However, the householder age had a negative effect on farmer household assets, which may be due to a decrease in the willingness to accept new technology and machinery as people get older. Zhang et al. (2020) pointed out that age was an important factor affecting the popularization and application of new machinery [35].

**Table 4.** The baseline regression results of farmer household income.

| Explanatory Variables | Coefficient | Standard Error | T Value | *p* Value |
|---|---|---|---|---|
| *PC* | 0.850 *** | 0.207 | 4.110 | 0.000 |
| *social* | 0.896 * | 0.464 | 1.930 | 0.054 |
| *human* | 1.204 | 0.864 | 1.390 | 0.165 |
| *land* | 0.001 *** | 0.000 | 6.600 | 0.000 |
| *labor* | 0.025 | 0.089 | 0.280 | 0.780 |
| *age* | −0.006 | 0.012 | −0.510 | 0.612 |
| *education* | −0.072 | 0.146 | −0.500 | 0.621 |
| *job* | −0.430 ** | 0.164 | −2.620 | 0.009 |
| *Constant* | 7.842 *** | 0.980 | 8.000 | 0.000 |
| *PseudoR$^2$* | 0.326 | | | |
| F value | 12.441 | | | |

Note: *, **, and *** represent 10%, 5%, and 1% significance, respectively.

**Table 5.** The baseline regression results of farmer household assets.

| Explanatory Variables | Coefficient | Standard Error | T Value | *p* Value |
|---|---|---|---|---|
| *PC* | 0.603 *** | 0.169 | 3.570 | 0.000 |
| *social* | 0.731 * | 0.404 | 1.810 | 0.072 |
| *human* | 0.414 | 0.717 | 0.580 | 0.564 |
| *land* | 0.001 *** | 0.000 | 7.490 | 0.000 |
| *labor* | 0.388 *** | 0.082 | 4.730 | 0.000 |
| *age* | −0.032 *** | 0.010 | −3.040 | 0.003 |
| *education* | −0.009 | 0.117 | −0.080 | 0.937 |
| *job* | −0.009 | 0.129 | −0.070 | 0.946 |
| *Constant* | 10.842 *** | 0.831 | 13.040 | 0.000 |
| *PseudoR$^2$* | 0.326 | | | |
| F value | 22.180 | | | |

Note: * and *** represent 10% and 1% significance, respectively.

### 4.2. Propensity Score Matching Analysis

In order to achieve sample matching, this paper selected the Logistic model to estimate the propensity score, referring to the research of Ma and Abdulai (2017), Meng et al. (2020), and Hoang (2021) [17,24,32], and sought the farmer households without cooperative membership, but that had similar economic characteristics with farmer households with cooperative membership for analysis. The model included the variables of farmer household social capital, farmer household human capital, farmer household land area under operation, farmer household labor scale, householder age, the education level of the householder, and migrant work experience of the householder.

Tables 6 and 7 show the results of propensity score matching analysis on farmer household income and assets, where the chi$^2$ values were 40.18 and 40.64, respectively, and the probability of being less than the *p* value for both was 0. Therefore, the null hypothesis was rejected, indicating that the model had a high level of fitting, while the independent variables greatly explained the dependent variable. Additionally, the consistent results presented in Tables 5 and 7 also showed that the higher the accumulation of farmer household social capital and human capital, the more farmers were inclined to participate in cooperatives. He et al. (2022) obtained a similar conclusion [36]. Additionally, compared with householders without migrant work experiences, householders who had been migrant workers were more likely to make the decision to participate in cooperatives.

**Table 6.** The results of propensity score matching analysis of farmer household income.

| Explanatory Variables | Coefficient | Standard Error |
|---|---|---|
| *social* | 3.688 *** | 0.990 |
| *human* | 4.345 * | 2.524 |
| *land* | 0.002 | 0.002 |
| *labor* | −0.075 | 0.178 |
| *age* | −0.008 | 0.032 |
| *education* | −0.376 | 0.333 |
| *job* | −0.711 ** | 0.348 |
| N | 248 | |
| *pseudoR$^2$* | 0.1364 | |
| *chi2* | 40.18 | |
| *Prob > chi2* | 0.000 | |

Note: *, **, and *** represent 10%, 5%, and 1% significance, respectively.

**Table 7.** The results of propensity score matching analysis of farmer household assets.

| Explanatory Variables | Coefficient | Standard Error |
|---|---|---|
| *social* | 3.681 *** | 0.988 |
| *human* | 4.367 * | 2.525 |
| *land* | 0.002 | 0.002 |
| *labor* | −0.077 | 0.178 |
| *age* | −0.009 | 0.032 |
| *education* | −0.379 | 0.333 |
| *job* | −0.674 ** | 0.345 |
| N | 248 | |
| *pseudoR$^2$* | 0.1368 | |
| *chi2* | 40.63 | |
| *Prob > chi2* | 0.000 | |

Note: *, **, and *** represent 10%, 5%, and 1% significance, respectively.

An important assumption in propensity score matching analysis was the assumption of balance, which required that there were no systematic differences in the matching variables between the treatment group and the control group after matching. In order to ensure the robustness of results, the balance test of propensity score matching was further tested by the methods of Psmatch2 and Pstest based on the existing research of Ma and Abdulai (2017) [17].

Tables 8 and 9 show consistent results; the deviation values of variables were all lower than 10%, indicating that the deviation was within the acceptable range. At the same time, the *p* values of all variables were greater than 10%, accepting the null hypothesis, indicating that after matching, there was no significant difference in the variables of economic characteristics between farmer households with cooperative membership and those without. The propensity score matching passed the balance test, and the matching effect was good.

**Table 8.** The results of the balance test of propensity score matching for farmer household income.

| Variables | Mean | | Deviation/% | T Value | *p* Value |
|---|---|---|---|---|---|
| | Participating | Non-Participating | | | |
| *social* | 0.631 | 0.621 | 6.30 | 0.390 | 0.699 |
| *human* | 0.552 | 0.550 | 1.80 | 0.120 | 0.908 |
| *land* | 16.414 | 31.010 | −2.20 | −1.770 | 0.179 |
| *labor* | 2.281 | 2.418 | −11.00 | −0.840 | 0.403 |
| *age* | 55.031 | 54.910 | 1.30 | 0.080 | 0.938 |
| *education* | 1.938 | 1.910 | 3.70 | 0.210 | 0.832 |
| *job* | 0.313 | 0.297 | 3.30 | 0.190 | 0.849 |

**Table 9.** The results of the balance test of propensity score matching for farmer household assets.

| Variables | Mean | | Deviation/% | T Value | *p* Value |
|---|---|---|---|---|---|
| | Participating | Non-Participating | | | |
| *social* | 0.635 | 0.625 | 5.9 | 0.37 | 0.712 |
| *human* | 0.557 | 0.549 | 5.2 | 0.35 | 0.73 |
| *land* | 16.394 | 32.091 | −2.4 | −1.87 | 0.064 |
| *labor* | 2.273 | 2.401 | −10.0 | −0.79 | 0.433 |
| *age* | 54.667 | 54.765 | −1.0 | −0.06 | 0.951 |
| *education* | 1.939 | 1.860 | 9.7 | 0.64 | 0.522 |
| *job* | 0.318 | 0.311 | 1.6 | 0.09 | 0.926 |

### 4.3. Average Treatment Effect Analysis

After propensity score matching, the ATT results of the impact of cooperative membership on farmer household income and assets are shown in Tables 10 and 11. It can be seen that the T value estimated by the four PSM methods of NNM, LLR, KM, and RM were all less than the results of baseline regression, and the overall effect was significant at a 1% level, which further verified the conclusion that cooperative membership can improve farmer household income and assets greatly, improvements that reached 7.04% and 4.19%, respectively.

**Table 10.** The ATT results of the impact of cooperative membership on farmer household income.

| Methods | Treatment Group | Control Group | ATT | Standard Error | T Value | △ (%) |
|---|---|---|---|---|---|---|
| NNM | 9.239 | 8.658 | 0.581 | 0.229 | 2.54 *** | 6.71% |
| LLR | 9.239 | 8.583 | 0.656 | 0.261 | 2.51 *** | 7.64% |
| KM | 9.239 | 8.618 | 0.621 | 0.216 | 2.88 *** | 7.21% |
| RM | 9.217 | 8.645 | 0.571 | 0.232 | 2.47 *** | 6.62% |
| Average | 9.234 | 8.626 | 0.607 | | | 7.04% |

Note: *** represent 1% significance.

**Table 11.** The ATT results of the impact of cooperative membership on farmer household assets.

| Methods | Treatment Group | Control Group | ATT | Standard Error | T Value | △ (%) |
|---|---|---|---|---|---|---|
| NNM | 11.187 | 10.742 | 0.444 | 0.199 | 2.23 *** | 4.14% |
| LLR | 11.187 | 10.696 | 0.491 | 0.250 | 1.96 *** | 4.49% |
| KM | 11.187 | 10.737 | 0.449 | 0.185 | 2.42 *** | 4.19% |
| RM | 11.192 | 10.779 | 0.413 | 0.204 | 2.02 *** | 3.83% |
| Average | 11.188 | 10.739 | 0.449 | | | 4.19% |

Note: *** represent 1% significance.

## 5. Conclusions and Discussion

In this paper, we selected the estimation method of propensity score matching (PSM) to quantitatively analyze the impact of membership in herbal medicine planting cooperatives on farmer household income and assets on the basis of an in-depth survey of the study area. We found that, firstly, cooperative membership can greatly improve farmer household income and assets, the migrant work experience of householders had a positive effect on farmer household income, the farmer household labor scale had a positive effect on farmer household assets, and the householder age had a negative effect on farmer household assets. Secondly, the higher the accumulation of farmer household social capital and human capital, the more farmers were inclined to participate in cooperatives. And compared with householders without migrant work experiences, householders who had been migrant workers were more likely to make the decision to participate in cooperatives. Finally, the

ATT results further verified the conclusion that cooperative membership can improve farmer household income and assets greatly, improvements that reached 7.04% and 4.19%, respectively.

It can be seen from the results that cooperative membership can greatly improve the income and assets of farmers. Here, the leading role of industrial development cannot be ignored. A large amount of research has verified the basic role of industries, especially the characteristic industries relying on regional resource endowment in economic development and farmers' income increase [7,37,38]. The herbal medicine industry selected in this paper was the main characteristic industry in Liaoning province. The existing varieties of herbal medicine mainly include understory ginseng, codonopsis, schisandra chinensis, asarum, gentian, acanthopanax, astragalus, etc. Huairen county in this region was named "the hometown of China's acanthopanax and understory ginseng", and "Fushun's schisandra chinensis" and "Fushun's understory ginseng" were certified as national geographic indications. Gentian and asarum occupied 80% and 70% of the national market sales, respectively.

In this context, cooperatives had become an important carrier of rural industrial prosperity, and only the coordinated development of cooperatives and industries can better serve rural revitalization. However, through survey and investigation, it was found that with the strong support of government departments at all levels, cooperatives in this region have developed rapidly, but there were still many problems, which made it difficult to meet the requirements of rural revitalization and agricultural and rural modernization in the new development stage. Overall, it mainly included five aspects:

Firstly, the development of quantity and quality was inconsistent. Despite the rapid development of cooperatives in recent years, their scale development and quality improvement were not synchronized, and the problem of "small, scattered, weak and empty" still existed. Survey data showed that the number of cooperatives in real operation was less than one-third of the total. Additionally, cooperatives with registered trademarks and quality certification of agricultural products only accounted for 4.33% and 1.32% of the total number in this region, and there were problems such as lagging brand building, standardization of agricultural products, and insufficient deep processing.

Secondly, the driving ability was insufficient. By the end of 2021, enterprise-led cooperatives accounted for less than 5% of the total in this region, and only 12.15% and 8.03% of cooperatives that uniformly organized the sale of agricultural products and purchased agricultural production inputs reached more than 80%. Overall, the radiation driving ability of cooperatives also needs to be improved.

Thirdly, the operating mechanism was irregular. On the one hand, the organizational structure was not perfect, and the internal interests of cooperatives were loosely connected. On the other hand, the financial management mechanism was not reasonable enough. Additionally, the cooperative surplus distribution was not standard enough. Most cooperatives mainly used price rebates instead of surplus distribution. In 2021, the number of cooperatives with surplus distribution returned by transaction volume was 4092, among which 2934 cooperatives returned more than 60%, accounting for only 5.23% of the total.

Fourthly, the support policy was inaccurate. In 2021, the total amount of government support funds in this region increased by 4.26% year-on-year, but only 3.68% of cooperatives received financial support funds, and less than 8% of the total undertook national agriculture-related projects. Additionally, although agriculture-related financial institutions had increased their credit support for "agriculture, rural areas and farmers", most financing products had certain limitations in terms of quota, use, and time, and problems such as difficult and expensive financing still existed.

Finally, the guidance service was inadequate. After the institutional reform in 2018, the guidance service work of cooperatives originally undertaken by agricultural economic stations (offices) was assigned to agricultural and rural administrative departments. However, the full-time staff of agricultural and rural administrative departments at the county

level in this region was generally no more than 3, resulting in insufficient guidance service work for cooperatives.

The political implications of this research include the following aspects: Firstly, innovating cooperation mechanisms and enriching joint forms by exploring different modes of cooperation among farmers using resource elements such as the contracted forestland and encouraging cooperatives closely tied to the industry to reorganize resources through mergers and amalgamations on a voluntary basis. Secondly, strengthening driving ability and stimulating endogenous driving forces through developing rural industries, and encouraging cooperatives to build specialized villages and towns with high quality, good benefits, and obvious advantages in leading industries, relying on local resource endowments, so as to form a development pattern with strong competitiveness, distinctive characteristics, and appropriate scale. Thirdly, enhancing driving capacity and stimulating internal driving forces by improving the organizational structure, standardizing financial management, and managing income distribution. Fourthly, strengthening system construction and improving standard level through increasing support for fiscal projects, innovating financial services, and focusing on digital empowerment. Finally, enhancing guidance services and strengthening institutional guarantees by establishing and improving a comprehensive and coordinated working mechanism, strengthening talent support to accelerate the development of basic institutions.

**Author Contributions:** Conceptualization, J.W. and. Z.Z.; methodology, Z.Z.; writing—original draft preparation, J.W.; writing—review and editing, L.G.; supervision, J.W. All authors have read and agreed to the published version of the manuscript.

**Funding:** This study was financially supported by the Youth Project of Liaoning Provincial Philosophy and Social Science Planning Fund (L21CJY012); the Commissioned Project of Young Talents Training Object of Philosophy and Social Sciences in Liaoning Province in 2022 (2022lslqnrcwtkt-34); and Fundamental Research Funds for the Provincial Universities of Zhejiang (22FR013).

**Institutional Review Board Statement:** Not applicable.

**Informed Consent Statement:** Not applicable.

**Data Availability Statement:** Not applicable.

**Conflicts of Interest:** The authors declare no conflict of interest.

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
