# Peer review of "Research on the Impact of Cooperative Membership on Forest Farmer Household Income and Assets—Case Study from Liaoning Herbal Medicine Planting Cooperatives, China"

_forests, doi:10.3390/f14091725_

Round 1
Reviewer 1 Report
The paper is written in a quality academic style and the methodology is understandable. The subject of the article is otherwise very current and especially important in modern society because natural resources are very important for humanity. Natural assets such as forests and plants can be renewed indefinitely, but their rate of regeneration must be taken into account. If the rate of exploitation of these natural resources is higher than their time rate of renewal (regeneration), the natural resources can be completely exhausted. The reviewer suggests that the paper also includes the sustainable dimension of development - the ecological dimension, i.e. that the authors point out that the association of farmers into cooperatives is also very good for the sustainability of natural resources, i.e. of the plant world. In conclusion, the authors pointed out that cooperatives are good because they promote the development of rural areas. The reviewer suggests that a couple of sentences about the sustainable management of natural resources be included in the paper.
The reviewer also suggests that at least five more references related to the management of forest and plant resources be included in the paper.

Author Response
Response to Comments from Reviewer 1
Reviewer:
The paper is written in a quality academic style and the methodology is understandable. The subject of the article is otherwise very current and especially important in modern society because natural resources are very important for humanity. Natural assets such as forests and plants can be renewed indefinitely, but their rate of regeneration must be taken into account. If the rate of exploitation of these natural resources is higher than their time rate of renewal (regeneration), the natural resources can be completely exhausted. The reviewer suggests that the paper also includes the sustainable dimension of development - the ecological dimension, i.e. that the authors point out that the association of farmers into cooperatives is also very good for the sustainability of natural resources, i.e. of the plant world. In conclusion, the authors pointed out that cooperatives are good because they promote the development of rural areas. The reviewer suggests that a couple of sentences about the sustainable management of natural resources be included in the paper.The reviewer also suggests that at least five more references related to the management of forest and plant resources be included in the paper.
Reviewer(1):
The authors should specify the research in terms of sustainable development, that is, they should also include an ecological dimension in forest management. Is increasing the income and property of forest farmers really the basis for achieving sustainable development in forestry?
Response:
Thanks for your kindly suggestion. During the process of investigation, we found that the two-way selection process between farmers and cooperatives in forest areas does not focus on the farmers' planting mode (planting in bare land or under forest), so this case cannot explain the role of herbal medicine cooperatives in forest resource management. Therefore, we only focus on the sustainability in economic way.
Reviewer(2):
How to solve a specific gap in this area? The topic of the paper is very current because it deals with the research of natural resources, which certainly includes forest resources, i.e. plant resource
Response:
Thanks for your suggestion. Since the varieties and techniques of planting Chinese herbal medicines in the study area are relatively uniform at present, the difference only exists in the understory planting and bare land planting, but from the planting, so the consideration of natural resources in this study only includes land.
Reviewer(3):
In terms of sustainability, it is necessary to mention the component of the renewal of natural funds, i.e. the rate of regeneration. Given that the paper is based more on the economic component, how much is taken into account in agricultural cooperatives in China about the rate of regeneration of natural funds of plants and forests?.
Response:
Thanks for your suggestion. According to our research, the cooperatives in the research area are still in the primary stage of economic cooperation organizations, and this part has not been involved yet. Therefore, we also put forward considerations on this issue in the conclusion and discussion sections.
Reviewer(4):
What further controls should be considered? Is monitoring carried out and who is monitoring the cooperatives that manage forest funds? Does the state implement appropriate control measures?
Reviewer(5):
The concluding remarks are adequate to the topic and the results of the work, except that it should also refer to the sustainable component of the management of forest resources, as well as to the government's assistance in implementing adequate forestry incentive measures.
Response:
Thanks for your suggestion. We have revised our discussion part to talk about this issue. The specific revision please see the original article.
Reviewer(6):
The references in the paper are adequate, but it is suggested that the authors also refer to the research of European authors.
Response:
Thanks for your suggestion. We have found very useful reference from European authors. The specific references are as following
- Rois-Díaz M, Lovric N, Lovric M, et al. Farmers’ reasoning behind the uptake of agroforestry practices: evidence from multiple case-studies across Europe[J]. Agroforestry Systems, 2018, 92: 811-828.
- Ficko A, Lidestav G, Dhubháin Á N, et al. European private forest owner typologies: A review of methods and use[J]. Forest Policy and Economics, 2019, 99: 21-31..
Reviewer(7):
The tables and figures in the paper are numbered, and as for the applied research methodology, it is necessary to include one more reviewer - a statistician - in the review process.
Response:
Thanks for your suggestion. We have double-checked the tables and figures in the paper to make sure they were correct.

Reviewer 2 Report
The results of this empirical study on the impact of forest farmers’ membership in herbal medicine planting cooperatives on their household income are interesting and new.
I have some remarks:
1. The title of the article should indicate the case of this study, i.e. forest farmers' membership in the herbal medicine planting cooperatives, in a mountainous forest area of Liaoning province of China. Why this is important - firstly, the People's Republic of China “promotes the development of farmer specialized cooperatives through financial support, preferential taxation, support for finance, science and technology, personnel, and guidance of industrial policies and other measures", which encourages the creation of cooperatives and the joining of forest farmers in cooperatives; secondly, this support has an impact on cooperatives' performance and/or costs, and the latter on household incomes; thirdly, in a mountainous area/region, the conditions of medicinal herb cultivation, raw material logistics, and market conditions are/may be different from plain areas. It is likely that in another country (for example, where the government does not support cooperatives), in another region (for example, in non-mountainous areas), in another forest industry (for example, logging), the results and conclusions of such a study would be different than those presented in this article.
2. The title of the article, the description of the research methodology, and the described results of the empirical research (survey of forest farmers) focus on the assessment of the impact of forest farmers' participation in the cooperative on the income of their households. Thus, the purpose of this study is to determine the effect of forest farmers (engaged in the cultivation of medicinal herbs) participation in a specialised cooperative on their household income. Therefore, the “cooperatives which relied on the characteristic industrial of herbal medicine planting in the eastern mountainous forest area of Liaoning province” themselves are not the “research object”, as stated in the introduction; and the text about herbal medicine industry cooperatives (e.g. problems of the current development of cooperatives in the forestry area, innovating cooperation mechanisms, cooperatives organizational structure, etc.) presented in the summary of the article, the discussion section and the conclusions of the cooperative does not correspond to the title of the article. Consequently, it is necessary to either change the title of the article and supplement the research methodology and the description of the results, or remove the text about the cooperative situation and their other issues, which are not related to the title of the article and the described research methodology, from the article.
3. In order to make the article clear to all readers, and to be able to replicate it for other researchers, the article must contain a complete and detailed description of definitions (measures) of research variables such as farmer household income (i.e. is it annual income or per season, in what currency?), farmer household social capital (social) and farmer household human capital (human) (what specific principal components were included in the factor analysis score in terms of both variables?), and farmer household land area under operation (is it all land used or just for growing herbs?) (Table 2).
4. It is unclear why sample N=248 was used for analysis (Table 2), although the text states that 511 valid questionnaires were obtained, with an effective rate of 95.93%. It is also unclear what the study population was in the mountainous forest area of Liaoning province.
5. The discussion section should primarily focus on the impact of forest farmers' participation in the cooperative on their household income. Meanwhile, the discussion prepared by the authors is completely unrelated to the empirical research results described in Chapter 4.
Author Response
Response to Comments from Reviewer
Reviewer:
The results of this empirical study on the impact of forest farmers’ membership in herbal medicine planting cooperatives on their household income are interesting and new.
I have some remarks:
Reviewer(1):
The title of the article should indicate the case of this study, i.e. forest farmers' membership in the herbal medicine planting cooperatives, in a mountainous forest area of Liaoning province of China. Why this is important - firstly, the People's Republic of China “promotes the development of farmer specialized cooperatives through financial support, preferential taxation, support for finance, science and technology, personnel, and guidance of industrial policies and other measures", which encourages the creation of cooperatives and the joining of forest farmers in cooperatives; secondly, this support has an impact on cooperatives' performance and/or costs, and the latter on household incomes; thirdly, in a mountainous area/region, the conditions of medicinal herb cultivation, raw material logistics, and market conditions are/may be different from plain areas. It is likely that in another country (for example, where the government does not support cooperatives), in another region (for example, in non-mountainous areas), in another forest industry (for example, logging), the results and conclusions of such a study would be different than those presented in this article.
Response:
Thanks for your kindly suggestion. We have changed our research title as “Research on the impact of cooperative membership on forest farmer household income and assets —— case study from Liaoning herbal medicine planting cooperatives, China”.
Reviewer(2):
The title of the article, the description of the research methodology, and the described results of the empirical research (survey of forest farmers) focus on the assessment of the impact of forest farmers' participation in the cooperative on the income of their households. Thus, the purpose of this study is to determine the effect of forest farmers (engaged in the cultivation of medicinal herbs) participation in a specialised cooperative on their household income. Therefore, the “cooperatives which relied on the characteristic industrial of herbal medicine planting in the eastern mountainous forest area of Liaoning province” themselves are not the “research object”, as stated in the introduction; and the text about herbal medicine industry cooperatives (e.g. problems of the current development of cooperatives in the forestry area, innovating cooperation mechanisms, cooperatives organizational structure, etc.) presented in the summary of the article, the discussion section and the conclusions of the cooperative does not correspond to the title of the article. Consequently, it is necessary to either change the title of the article and supplement the research methodology and the description of the results, or remove the text about the cooperative situation and their other issues, which are not related to the title of the article and the described research methodology, from the article.
Response:
Thanks for your suggestion. We have revised our research object. The revised statement is as follows:
“This paper selected the farmer households that planting herbal medicine in the eastern mountainous forest area of Liaoning province as research object. Empirical tested the im-pact of membership in herbal medicine planting cooperatives on forest farmer household income and assets through the combination of theory and demonstration. Explored the po-tential impact of cooperatives on continuously promoting the increase of household income and guaranteeing the stable and sustainable development of forestry area, in order to obtain new supporting evidence.”
Reviewer(3):
In order to make the article clear to all readers, and to be able to replicate it for other researchers, the article must contain a complete and detailed description of definitions (measures) of research variables such as farmer household income (i.e. is it annual income or per season, in what currency?), farmer household social capital (social) and farmer household human capital (human) (what specific principal components were included in the factor analysis score in terms of both variables?), and farmer household land area under operation (is it all land used or just for growing herbs?) (Table 2).
Response:
Thanks for your suggestion. We have modified this part according the research results. The revised statement is as follows:
“Table 1. The definitions of variables.
Variables |
Definitions |
Participating cooperatives (PC) |
Participating (as treatment group)=1; Non- participating (as control group)=0. |
Farmer household income (income) |
The logarithmic of the total income related to herbal medicine planting and processing of farmer household in 2021 (in CNY). |
Farmer household assets (assets) |
The logarithmic of the valuation of agricultural operating fixed assets owned by farmer household in 2021 (in CNY) |
Farmer household social capital (social) |
Principal component factor analysis score. |
Farmer household human capital (human) |
Principal component factor analysis score. |
Farmer household land area under operation (land) |
The total area of land plating herbal medicine by the farmer household. |
Farmer household labor scale (Labor) |
The number of labor forces participated in herbal medicine planting and processing. |
Householder age (age) |
- |
The education level of householder (education) |
Primary school=1; junior high school=2; high school=3; college=4. |
The migrant work experience of householder (migrant) |
Yes=1; No=2. |
Tabl2 Composition of farmer household social capital and human capital
Variables |
Dimension |
Definitions |
Farmer household social capital (social) |
Social network |
Relationship with relatives and friends |
Relationship with village leaders |
||
Social trust |
The number of people you can trust in your village |
|
The degree of trust in the village committee |
||
Possibility of finding something lost in the village |
||
Social prestige |
The Importance of Individual Opinions in Village Collective Decision-Making |
|
Probability of others asking you for help when they are in trouble |
||
Frequency of mentoring others |
||
Frequency of resolving conflicts in the village |
||
Social participation |
Willingness to participate in village collective activities |
|
Farmer household human capital (human) |
Knowledge |
Average education level of the household workforce |
Capacity |
Average age of the household workforce |
|
Average health level of household |
Reviewer(4):
It is unclear why sample N=248 was used for analysis (Table 2), although the text states that 511 valid questionnaires were obtained, with an effective rate of 95.93%. It is also unclear what the study population was in the mountainous forest area of Liaoning province.
Response:
Thanks for your suggestion. The overall data we have collected is 511, however, some of our samples are not involving with herbal medicine planting. Thus, we finally choose 248 farmers who planted herbal medicine. To make it clearer, we have revised the statement as follows:
“In this paper, 248 farmers who growing herbal medicine in forest area were selected as the specific analysis objects.”
Reviewer(5):
The discussion section should primarily focus on the impact of forest farmers' participation in the cooperative on their household income. Meanwhile, the discussion prepared by the authors is completely unrelated to the empirical research results described in Chapter 4.
Response:
Thanks for your suggestion. We have modified this part and merged the conclusion part and discussion part, the specific revision please see the original article.
